# Global distribution of particle phase state in atmospheric secondary organic aerosols

Manabu Shiraiwa[1,2], Ying Li[2,3,4], Alexandra P. Tsimpidi[5], Vlassis A. Karydis[5], Thomas Berkemeier[2,6], Spyros N. Pandis[7], Jos Lelieveld[5,8], Thomas Koop[9] & Ulrich Pöschl[2]

Secondary organic aerosols (SOA) are a large source of uncertainty in our current understanding of climate change and air pollution. The phase state of SOA is important for quantifying their effects on climate and air quality, but its global distribution is poorly characterized. We developed a method to estimate glass transition temperatures based on the molar mass and molecular O:C ratio of SOA components, and we used the global chemistry climate model EMAC with the organic aerosol module ORACLE to predict the phase state of atmospheric SOA. For the planetary boundary layer, global simulations indicate that SOA are mostly liquid in tropical and polar air with high relative humidity, semi-solid in the mid-latitudes and solid over dry lands. We find that in the middle and upper troposphere SOA should be mostly in a glassy solid phase state. Thus, slow diffusion of water, oxidants and organic molecules could kinetically limit gas–particle interactions of SOA in the free and upper troposphere, promote ice nucleation and facilitate long-range transport of reactive and toxic organic pollutants embedded in SOA.

[1] Department of Chemistry, University of California, Irvine, California 92697-2025, USA. [2] Multiphase Chemistry Department, Max Planck Institute for Chemistry, 55128 Mainz, Germany. [3] State Key Laboratory of Atmospheric Boundary Layer Physics and Atmospheric Chemistry (LAPC), Institute of Atmospheric Physics, Chinese Academy of Sciences, Beijing 100029, China. [4] Center for Regional Environmental Research, National Institute for Environmental Studies, Tsukuba 305-0053, Japan. [5] Atmospheric Chemistry Department, Max Planck Institute for Chemistry, 55128 Mainz, Germany. [6] School of Chemical and Biomolecular Engineering, Georgia Institute of Technology, Atlanta, Georgia 30332-0100, USA. [7] Department of Chemical Engineering, University of Patras, Patras 265 04, Greece. [8] The Cyprus Institute, Nicosia 1645, Cyprus. [9] Faculty of Chemistry, Bielefeld University, 33615 Bielefeld, Germany. Correspondence and requests for materials should be addressed to M.S. (email: m.shiraiwa@uci.edu).

Secondary organic aerosols (SOA) account for a large fraction of submicron particles in the atmosphere, impacting clouds, climate, air quality and public health[1,2]. SOA particles influence climate by scattering sunlight and serving as nuclei for cloud droplets and ice crystals. SOA represent a large source of uncertainty in current understanding of global climate change and air pollution[2,3]. Traditionally, SOA particles have been assumed to be homogeneous and well-mixed liquids. Recent laboratory experiments as well as atmospheric measurements, however, have demonstrated that they can occur in amorphous solid or semi-solid phase states depending on chemical composition, relative humidity (RH) and temperature[4–6]. The particle phase state is crucial for various atmospheric gas–particle interactions[7], including SOA formation and partitioning[8–10], heterogeneous and multiphase reactions[11,12] and ice nucleation[13–16]. The glass transition temperature ($T_g$) characterizes the non-equilibrium phase change from a glassy solid state to a more pliable semi-solid state as the temperature increases[5]. It is important to know the SOA phase state in multicomponent atmospheric particles for better quantification of aerosol effects on climate and air quality; but little information is available on the spatial distribution of the particle phase state of SOA on regional and global scales.

Here we developed a method to estimate $T_g$ of SOA components and provided a first estimate of the global distribution of the SOA phase state by applying the global chemistry climate model. In the planetary boundary layer, we found that SOA are liquid in tropical and polar air with high RH, semi-solid in the mid-latitudes and solid over dry lands. In the middle and upper troposphere, SOA should be mostly in a glassy state, which may promote ice nucleation and facilitate long-range transport of organic pollutants embedded in SOA.

## Results

**Molecular corridor and glass transition temperature.** Our analysis of SOA phase state builds on the molecular corridor approach[17,18], which is a two-dimensional framework of volatility and molar mass of SOA components constrained by boundary lines of low and high molecular O:C ratio. Figure 1a shows the interdependence of volatility, molar mass and $T_g$ for 654 SOA components composed of C, H and O formed upon oxidation of

biogenic precursors (isoprene, α-pinene, limonene, glyoxal) as well as anthropogenic precursors ($C_{12}$ alkanes)[17]. The upper and lower bounds of the corridor into which the data fall are given by two lines representing n-alkanes ($C_nH_{2n+2}$) with O:C = 0 and sugar alcohols ($C_nH_{2n+2}O_n$) with O:C = 1.

Figure 1b shows how $T_g$ depends primarily on the molar mass and secondarily on the O:C ratio of the organic molecules, as previously demonstrated by Koop et al.[5] and further illustrated in Supplementary Fig. 1a. Based on the available measurement data, we developed a new parameterization of $T_g$ as a function of molar mass and O:C ratio as detailed in the method section. Figure 1c shows that the $T_g$ values predicted with the new parameterization agree well with the $T_g$ values measured in experiments or estimated by the Boyer–Kauzmann rule (see Methods), with a correlation coefficient of 0.93. The $T_g$ of individual compounds can be predicted within ± 20 K, however, this uncertainty may be reduced to ± 3 K for multicomponent SOA mixtures under ideal mixing conditions.

The parameterization scheme has been implemented into the global chemistry climate model EMAC[19] that includes the organic aerosol module ORACLE, based on a computationally efficient description of primary and secondary organic aerosol sources, phase-partitioning and chemical evolution[20]. ORACLE uses the volatility basis set (VBS) framework[21] for distributing SOA oxidation products into four logarithmically spaced volatility bins with effective saturation concentrations $C^*$ of 1, 10, $10^2$ and $10^3$ μg m$^{-3}$, respectively[20]. In this work the values of molar mass and O:C ratio of oxidation products in different volatility bins were assigned based on molecular corridors[17,18] and previous studies (Supplementary Table 1). $T_g$ of dry SOA products in each volatility bin was calculated using the new parameterization. $T_g$ of atmospheric SOA mixed with water due to hygroscopic growth was estimated using the Gordon–Taylor approach[5] (see Methods). Sensitivity simulations show that variations of the key parameters including molar mass, hygroscopicity and the Gordon–Taylor constant lead to changes of $T_g$ within 15% (Supplementary Material).

The SOA phase state can be inferred using the inverse ambient temperature ($1/T$) scaled by the glass transition temperature of SOA, that is, $T_g/T$. When the ambient temperature is below $T_g$ (that is, $T_g/T \geq 1$), an amorphous particle behaves as a solid and kinetic limitations occur; in contrast, when $T_g/T < 1$, a particle

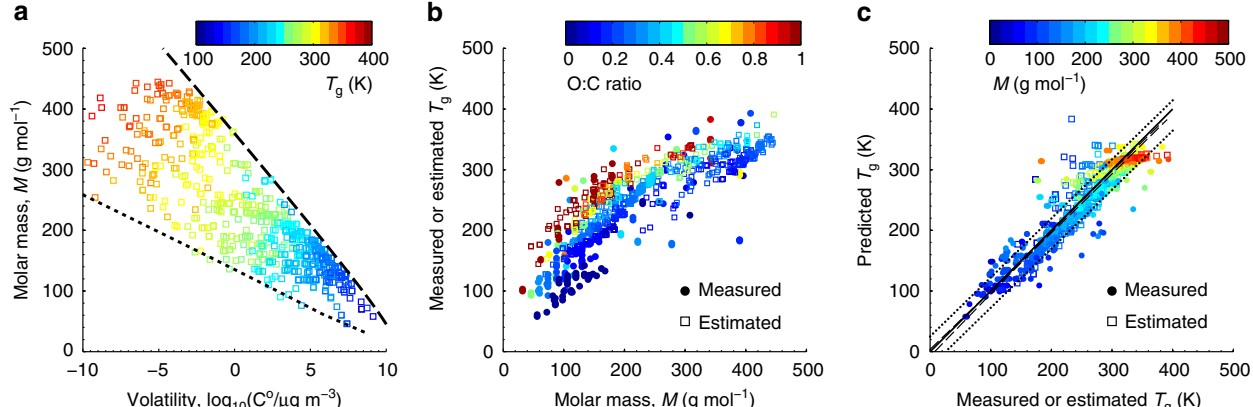

**Figure 1 | Characteristic relations between molecular properties and glass transition temperature of organic compounds.** (**a**) Molecular corridor of molar mass plotted against volatility of 654 SOA compounds[17] colour-coded by glass transition temperature ($T_g$) estimated with the Boyer–Kauzmann rule[5]. The upper dashed line indicates the low O:C bound of the molecular corridor (linear alkanes $C_nH_{2n+2}$ with O:C = 0), the lower dotted line indicates the high O:C bound (sugar alcohols $C_nH_{2n+2}O_n$ with O:C = 1). (**b**) Measured (circles) and estimated (squares) $T_g$ of organic compounds plotted against molar mass. Organic compounds with measured $T_g$ are from Koop et al.[5] and Dette et al.[51]. The markers are colour-coded by molecular O:C ratio. (**c**) Predicted $T_g$ using a parameterization developed in this study compared to measured (circles) and estimated (squares) $T_g$ with the Boyer–Kauzmann rule. The solid line shows 1:1 line and the dashed and dotted lines show 68% confidence and prediction bands, respectively.

exists in semi-solid or liquid states. The threshold between semi-solid and liquid states depends on the so-called fragility of the SOA compounds; in this study, we assume an average fragility value of various organic compounds[22], resulting in the threshold of $T_g/T \approx 0.8$.

Figure 2 shows global simulations of the annual average of $T_g/T$ at mean RH at the surface, 850 hPa ($\sim$1.35 km) and 500 hPa ($\sim$5.5 km), using simulated ambient RH and $T$ data for the years 2005–2009. These global patterns of the SOA phase state are strongly influenced by RH, as an increase leads to a significant decrease of $T_g$ due to hygroscopic growth of particles (Supplementary Figs 2,3,4). Near the Earth's surface, SOA is mostly liquid in the tropics—including the Amazon rainforest—and in polar air with high RH. This result is consistent with recent particle bounce measurements reporting that SOA particles over the Amazon basin are liquid at RH higher than 80% during both the wet and dry seasons[23].

**Global modelling of SOA phase state.** In mid-latitudes and semi-arid regions such as in the western US, Mexico, Europe, India and Australia, a semi-solid organic phase is predicted. This is in line with measurements showing that organic particles collected in California, Mexico City and Chile may have higher viscosities[24]. Marine SOA are also predicted to be semi-solid even at high RH, as they typically have a high molar mass and O:C ratio due to extended chemical aging. SOA particles are likely to be amorphous solids in arid regions of the dust belt (for example, Sahara, Arabian, Gobi) in North Africa, the Middle East and Central Asia with low RH. A glassy solid state is predicted over the boreal forest, for example, in Finland at RH of $\leq$30% (Supplementary Fig. 4), in agreement with observations based on particle bounce measurements[4], corroborating that SOA particles are amorphous solids.

Convective transport of SOA particles to higher altitudes leads to more frequent occurrence of solid or semi-solid phases as shown in Fig. 2. At 850 hPa solid particles prevail over most of the continents at low and mid latitudes, while particles remain semi-solid or liquid over the continents in the tropics and at high latitudes as well as over the oceans. After further uplifting to 500 hPa, corresponding to an average altitude of about 5.5 km, almost all SOA particles are expected to undergo transition into a glassy solid state. The occurrence of viscous states at low temperature is consistent with recent chamber experiments, showing that $\alpha$-pinene-derived SOA particles exist in a viscous state at low temperatures corresponding to the cirrus cloud altitude region of the free troposphere[25].

Figure 3a shows the mean vertical profiles of $T_g/T$ calculated for selected regions in the Amazon basin, Europe, East China, US, India and Sahara, as specified by black boxes in the top panel of Fig. 2. Over the Amazon basin, SOA particles are predicted to remain liquid or semi-solid up to 5 km and are solidified only above about 5 km. Amazonian-SOA particles have relatively low molar mass and O:C, due to the dominance of strong isoprene emissions and shorter chemical aging times, which in combination with relatively high temperature and RH results in low $T_g$. In contrast, an amorphous solid state is expected for SOA particles at near-surface altitudes over the Sahara, where RH is usually lower than $\sim$40% up to $\sim$8 km (Supplementary Fig. 5b). Mass concentrations of SOA over the Sahara are very low, being remote from sources, but SOA particles are highly aged due to chemical processing during long-range transport, and thus are expected to have higher molar mass and $T_g$. In other regions, SOA particles are expected to be liquid up to $\sim$2 km in the planetary boundary layer, above which they undergo a glass transition when the temperature becomes lower than $\sim$270 K (Supplementary Fig. 5a).

**Characteristic diffusion timescales.** As illustrated in Fig. 3b, SOA particles serve as nuclei for clouds and precipitation, sustaining the hydrological cycle[26]. Figure 3c shows characteristic diffusion timescales ($\tau_{cd}$) of water molecules in SOA particles with a diameter of 200 nm. These are of the order of microseconds at the Earth's surface and seconds at 850 hPa. These timescales are short, so that inhibition of activation of cloud condensation nuclei (CCN) is not expected, in agreement with previous particle trajectory modelling[27]. In contrast, the particle phase state is fundamental for different ice nucleation (IN) pathways: liquid particles can freeze homogeneously, whereas (semi-)solid

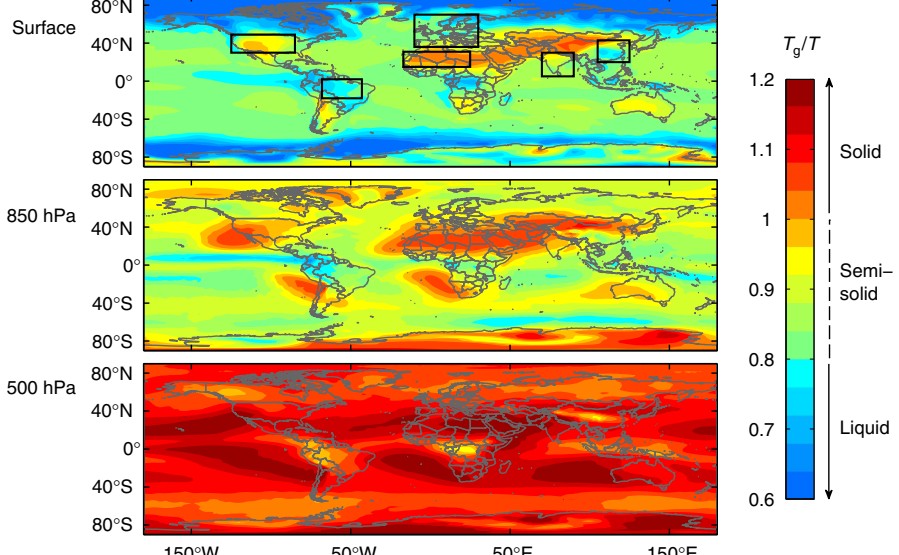

**Figure 2 | SOA phase state in the global atmosphere.** Modelled annual averages of the inverse ambient temperature ($1/T$) scaled by the glass transition temperature ($T_g$) of SOA ($T_g/T$) at the surface, 850 and 500 hPa, respectively, for the years 2005–2009. $T_g/T$ is an indicator of the particle phase state: $T_g/T \geq 1$, solid; $\sim$0.8 $< T_g/T <$1, semi-solid; $T_g/T \leq \sim$0.8, liquid. The black squares in the top panel indicate specific areas over the Amazon basin, US, Europe, Sahara, India and East China, respectively (Fig. 3a).

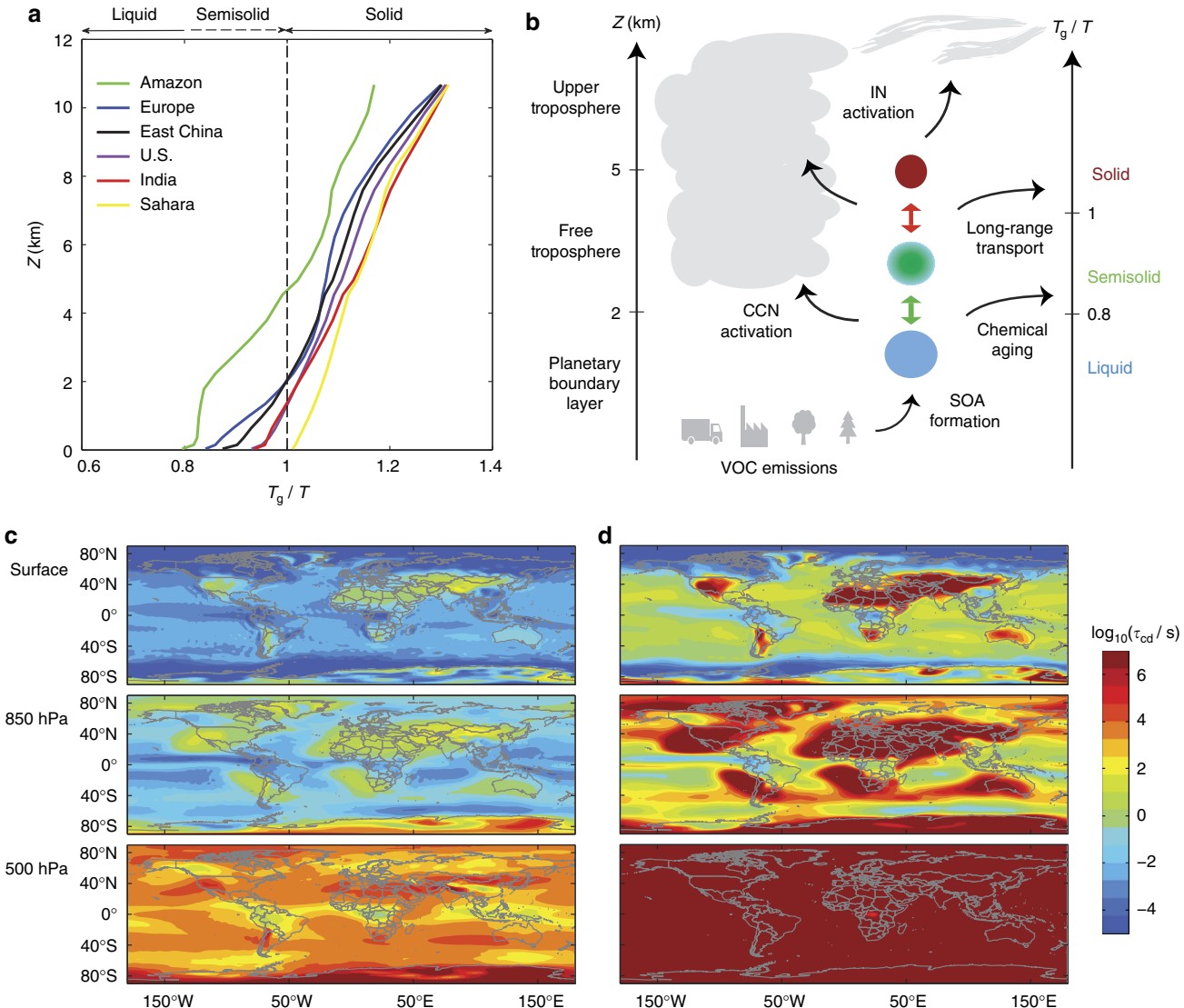

**Figure 3 | SOA phase state and atmospheric implications.** (**a**) Modelled mean vertical profiles of the inverse ambient temperature ($1/T$) relative to the glass transition temperature ($T_g$) of SOA ($T_g/T$) in specific areas over the Amazon basin, US, Europe, Sahara, India and East China as defined in Fig. 2. (**b**) Schematic of evolution of SOA phase state as a function of altitude and temperature, which has significant implications for CCN/IN activation, chemical aging and long-range transport. (**c,d**) Characteristic mixing timescales of (**c**) water and (**d**) organic molecules in SOA particles at the surface, 850 hPa and 500 hPa

particles can be expected to form ice crystals heterogeneously[14,16]. For example, at 500 hPa, timescales of water diffusion can be up to several hours. In that case, glassy states and solid/liquid core-shell morphologies can persist sufficiently long, so that heterogeneous ice nucleation in the deposition and immersion modes may dominate over homogeneous ice nucleation at lower temperatures ($< \sim 240$ K)[15].

Figure 3d shows global distributions of characteristic mixing timescales of organic molecules within the SOA matrix. They are generally several orders of magnitude larger than mixing timescales of water molecules, because bulk diffusivities of organic molecules are lower than those of water due to their larger molar mass[11]. Characteristic mixing times are shorter than minutes over oceans, tropics and high latitudes at the surface and 850 hPa, indicating that particles are homogeneously mixed and are likely to be in equilibrium between the gas and particulate phases. However, characteristic timescales are more than a day over dry regions at the surface, over most continental regions at 850 hPa and over the entire globe at 500 hPa, where particles are

practically 'frozen' and subject to strong kinetic limitations of bulk diffusion of organic molecules. In such conditions, kinetic partitioning[8–10] may be driving new particle formation and subsequent particle growth in the free troposphere[28].

## Discussion

These results have also important implications for long-range transport of persistent organic pollutants and polycyclic aromatic hydrocarbons. There are numerous observations of high concentrations of such hazardous and toxic compounds at remote sites such as the Arctic and the Antarctic[29]. This is unexpected, as chemical half-lives of these compounds against degradation by atmospheric oxidants such as $O_3$, OH and $NO_3$ have been estimated to be much shorter than the transport timescale. One plausible explanation arising from this study is that toxic compounds are embedded within glassy SOA matrices with low-bulk diffusivities and long mixing timescales, which can effectively shield them from chemical degradation by atmospheric

photo-oxidants, facilitating efficient long-range transport in the atmosphere[30–32].

The results of this study highlight that due to a strong dependence of the particle phase state on RH and ambient temperature, the latter parameters determine the spatiotemporal distributions of SOA phase state worldwide. Several important aspects should be further explored in dedicated studies, such as diel and seasonal variations of the phase state, dependence of phase state on particle size[33] and various anthropogenic and biogenic precursors, interactions with inorganic components in particles, and effects of extremely low-volatile organic compounds (ELVOCs)[34] as well as particle- and aqueous-phase chemistry[35]. These results indicate that further development of advanced and detailed formalisms for the SOA lifecycle description in atmospheric models, and the effects on climate, air quality and public health is required.

## Methods

**ORACLE module.** The organic aerosol module ORACLE considers the oxidation of alkanes, aromatics and isoprene by OH, and the oxidation of olefins and monoterpenes by $O_3$, OH and $NO_3$ (ref. 20). The global emission inventories of the considered SOA precursors are described in detail in Tsimpidi *et al.*[20]. The model system is applied to simulate the period 2005–2009 with a horizontal grid resolution of $1.875° \times 1.875°$ and 31 vertical layers extending to about 25 km altitude (10 hPa). The model system has been evaluated by comparisons to observations, showing good performance in reproducing the spatial distributions and mass concentrations of SOA[36].

The VBS approach assumes equilibrium partitioning. Ye *et al.*[37] conducted chamber experiments, and found that α-pinene SOA mixes relatively quickly above 40% RH at room temperature. Liu *et al.*[38] also observed a rapid partitioning of SOA compounds above 20–30% RH for anthropogenic SOA and for all RH values for biogenic SOA. These new studies strongly suggest that kinetic limitations in the bulk may not significantly affect SOA partitioning in the boundary layer, justifying the use of equilibrium partitioning in this part of the atmosphere. As shown in Supplementary Fig. 5c, in the current model, most SOA formation (around 95%) occurs in the boundary layer with conditions similar to the above experimental studies. Therefore, the apparent inconsistency may be relevant to only a small fraction (around 5%) of the SOA formation in this modelling study, and does not significantly affect our results and our conclusions. Application of a model such as the kinetic multi-layer model of gas–particle interactions (KM-GAP[39,40]) in both the interpretation of laboratory measurements and in the global model should be explored in future work, but is beyond the scope of the current study.

The ORACLE module simulates SOA components with effective saturation concentrations as low as $0.01\,\mu g\,m^{-3}$ at 298 K for SOA from semi-volatile and intermediate volatility compounds. Compounds of even lower volatility (low-VOCs and extremely low-VOCs)[34] are not simulated for reasons of computational efficiency and because the predicted OA mass concentration fields are relatively insensitive to their explicit simulation. The simulation of the extremely low volatility compounds is mostly relevant for the prediction of number concentrations due to their role in the formation and growth of new particles[41]. Nevertheless, these compounds are implicitly included in the corresponding lower volatility bin of the VBS during the fitting of the corresponding smog chamber results. The accuracy of this simplification has been evaluated in previous applications of ORACLE and the comparisons of its predictions with available field measurements[36].

**Global simulations of $T_g$.** The values of molar mass for anthropogenic and biogenic SOA compounds in different volatility bins were assigned based on molecular corridors as specified in Supplementary Table 1. The slopes of molecular corridors correspond to the increase in molar mass required to decrease volatility by one order of magnitude, $-dM/dlogC^0$, which ranges from $\sim 15\,g\,mol^{-1}$ for isoprene to $\sim 25\,g\,mol^{-1}$ for $C_{12}$ alkane[17]. According to the corridor slopes, the molar mass in every volatility bin of 1, 10, $10^2$ and $10^3\,\mu g\,m^{-3}$ was assigned for SOA oxidation products formed from VOCs. Note that the molar mass assigned for the volatility bin of $1\,\mu g\,m^{-3}$ is assumed to have relatively high molar mass based on molecular corridors to compensate for the fact that the current model does not consider lower volatility bins with higher molar mass. The assignment of O:C ratios for each volatility bin was based on previous studies[42,43]. Both molar mass and the O:C ratio were varied for sensitivity studies to investigate the effects of these parameters on global simulations of $T_g$.

In this work, we implicitly assume that SOA particles are externally mixed with inorganic compounds such as sulfate and nitrate, similar to the current treatment in most large-scale models. If they were internally mixed, phase separation of organic and inorganic compounds can be expected. Experimental[44,45] and modelling[46–48] studies as well as field observations of phase separated particles[49,50] demonstrate that a liquid–liquid phase separation is very likely to occur when

O:C < 0.5 for the organic fraction, but unlikely (that is, one mixed phase occurs) when O:C > 0.7. SOA predicted in this work usually show O:C < 0.7 (Supplementary Fig. 1); thus, an organic phase is likely to be phase separated from the inorganic phase. Nevertheless, phase partitioning of internal mixing of organic and inorganic compounds is subject to future studies.

Glass transition temperatures of SOA products in each volatility bin ($T_{g,i}$) were calculated using the following parameterization developed in this study, with the average molar mass and O:C ratio of SOA compounds in each volatility bin, assigned based on molecular corridors and previous studies.

$$T_{g,i} = A + BM + CM^2 + D(O:C) + EM(O:C) \tag{1}$$

The units of $T_{g,i}$ and $M$ are K and $g\,mol^{-1}$, respectively, while O:C is dimensionless. Values of coefficients ($A$, $B$, $C$, $D$, $E$) were obtained by fitting with multi-linear least squares analysis with 68% prediction and confidence intervals to the experimental $T_g$ of 179 CH and CHO compounds, resulting in $A = -21.57$ ($\pm 13.47$) (K), $B = 1.51$ ($\pm 0.14$) ($K\,mol\,g^{-1}$), $C = -1.7 \times 10^{-3}$ ($\pm 3.0 \times 10^{-4}$) ($K\,mol^2\,g^{-2}$), $D = 131.4$ ($\pm 16.01$) (K) and $E = -0.25$ ($\pm 0.085$) ($K\,mol\,g^{-1}$), respectively. $T_{g,i}$ is assumed to represent average glass transition temperature of a complex mixture of thousands of organic molecules in each volatility bin. Even though this is an approximation, it is a good starting point for global $T_g$ estimation, making use of the VBS and molecular corridor approach.

Equation (1) was also applied to predict $T_g$ of 654 SOA compounds[17] to compare with their estimated $T_g$ (Fig. 1c). The melting points ($T_m$) of these SOA compounds were calculated by the Estimation Programs Interface Suite software and $T_g$ was subsequently estimated using the Boyer–Kauzmann rule of $T_g = g \cdot T_m$ with $g = 0.7$ (see validation of this method in Supplementary Fig. 1b)[5]. As shown in Fig. 1c, equation (1) predicts $T_g$ well and some of the variations in Fig. 1c may be related in part to experimental uncertainties in the determination of $T_g$, originating from different experimental protocols, temperature calibration issues, differences in sample purity and treatment[5,51].

$T_g$ of mixtures of SOA compounds under dry conditions ($T_{g,org}$) were calculated using the Gordon–Taylor approach with a linear relationship[51]: $T_{g,org} = \sum w_i T_{g,i}$, where $w_i$ is the mass fraction of organic compound $i$, which can be derived using mass concentrations of SOA products simulated by the ORACLE module. This equation is a simplified form of the Gordon–Taylor equation that is valid when the Gordon–Taylor constants are equal to 1 (ref. 51). Under humid conditions, SOA particles take up water by hygroscopic growth in response to ambient RH. $T_g$ of organic–water mixtures can be simulated as well using the Gordon–Taylor equation[5]:

$$T_g(w_{org}) = \frac{(1 - w_{org})T_{g,w} + \frac{1}{k_{GT}} w_{org} T_{g,org}}{(1 - w_{org}) + \frac{1}{k_{GT}} w_{org}} \tag{2}$$

where $w_{org}$ is the mass fraction of organics, $T_{g,w}$ is the glass transition temperature of pure water (136 K) and $k_{GT}$ is the Gordon–Taylor constant which is assumed to be 2.5 ($\pm 1.0$) (refs 5,13).

The Gordon–Taylor approach has been applied and validated for a wide range of mixtures including organic solvent/polymer, water/polymer, molecular organics/molecular organics, as well as water/molecular organics mixtures[13,52,53]. For SOA compounds the approach has been validated for mixtures of the α-pinene oxidation products 3-MBTCA and pinonic acid[51] and we assume that it can be also applied to multicomponent mixtures. It has been pointed out previously that the Gordon–Taylor approach may fail in the case of adduct or complex formation[5], which, however, is more likely to occur in binary or ternary mixtures, where the two strongly interacting compounds occur at high mole fractions. However, adduct or complex formation is highly unlikely in multicomponent mixtures such as SOA with hundreds of compounds, because the individual mole fractions are very small. Thus, in multicomponent mixtures particular interactions between individual compounds are more likely to average out, thus favoring a mean-field type Gordon–Taylor approach.

The mass concentration of water ($m_{H2O}$) absorbed by SOA particles can be calculated using the effective hygroscopicity parameter ($\kappa$) as: $m_{H2O} = \frac{\kappa \rho_w m_{SOA}}{\rho_{SOA}\left(\frac{1}{a_w} - 1\right)}$ (ref. 54). The density of water ($\rho_w$) is $1\,g\,cm^{-3}$, the density of SOA particles ($\rho_{SOA}$) is assumed to be $1.4\,g\,cm^{-3}$ (ref. 55), $m_{SOA}$ is the simulated total mass concentrations of SOA and $a_w$ is the water activity calculated as $a_w = RH/100$. $\kappa$ is assumed to be 0.1 ($\pm 0.05$) based on field measurements[56–59], laboratory experiments[60–63] and global simulations[64].

Viscosity ($\eta$) can be estimated from the $T_g$-scaled Arrhenius plot of $\eta$ versus $T_g/T$ (ref. 22) as shown in Supplementary Fig. 6. This relation depends on the fragility parameter $D$: larger $D$ leads to Arrhenius (or strong) behaviour, while smaller $D$ leads to fragile behaviour. Typical $D$ values for organic compounds are in the range of $\sim 5$–20 (ref. 65). As $D$ values for SOA compounds are unknown, we assume $D = 10$ for the base case leading to the threshold of liquid and semi-solid at $T_g/T = 0.79$ ($\pm 0.1$). Bulk diffusivities of organic molecules (Supplementary Fig. 7) can be converted from viscosities assuming the Stokes–Einstein relation[11]. Using the derived bulk diffusivities, the characteristic timescales of bulk diffusion ($\tau_{cd}$) in a particle with diameter $d_p$ can be calculated as $\tau_{cd} = d_p^2/4\pi^2 D_b$ (ref. 11). The Stokes–Einstein relation may break down when $T_g$ is close to $T$ (ref. 66), leading to underestimation of bulk diffusivities, and hence, overestimation of characteristic timescales of the bulk diffusion of organic molecules.

**Water diffusivities and characteristic mixing timescales.** For the estimation of diffusivities of small molecules (for example, $H_2O$) diffusing through a (semi-)solid matrix, the Stokes–Einstein equation is not applicable[11,15,47,66,67]. Diffusion coefficients of water ($D_{H_2O}$) in SOA have been estimated using a semi-empirical method of Berkemeier et al.[15]. The method can estimate water diffusivities over a large range in $T$ and RH, when the glass transition temperature and hygroscopicity of a target organic mixture are known. This is achieved by extrapolating from a reference substance of sucrose[68] under the assumption that the target organic mixture is chemically similar to the reference substance and thus exhibits a similar Vogel–Fulcher–Tammann (VFT) parameter $B$ (ref. 22) in the VFT equation[15].

$$\log_{10}(D_{H_2O}) = A + \frac{B}{T - T_0} \qquad (3)$$

$A$ and $B$ for the VFT equation were directly adopted from Zobrist et al.[68] as below:

$$A = 3 + 0.175 \cdot \left(1 - 46.46\left(1 - a_{w,suc}\right)\right) \qquad (4)$$

$$B = 262.867 \cdot \left(1 + 10.53\left(1 - a_{w,suc}\right) - 0.3\left(1 - a_{w,suc}\right)^2\right) \qquad (5)$$

For determination of $T_{0,SOA}$ from $T_{0,suc}$, target (SOA) and reference (sucrose) substances are compared at identical mass fraction. Given the density of the organic material $\rho_{org}$ (g cm$^{-3}$) and hygroscopicity $\kappa$, the mass fraction of SOA can be calculated at a given RH[5].

$$w_{SOA} = \frac{\rho_{org}\left(1 - \frac{RH}{100}\right)}{\frac{RH}{100}\left(\kappa - \rho_{org} + \frac{\rho_{org}}{\frac{RH}{100}}\right)} \qquad (6)$$

The water activity in sucrose at the given SOA weight fraction can be calculated as[68]:

$$a_{w,suc} = \frac{1 + a \cdot w_{SOA}}{1 + b \cdot w_{SOA} + c \cdot w_{SOA}^2} + \left(T - T_{ref}\right)\left(d \cdot w_{SOA} + e \cdot w_{SOA}^2 + f \cdot w_{SOA}^3 + g \cdot w_{SOA}^4\right) \qquad (7)$$

where $a = -1$, $b = -0.99721$, $c = 0.13599$, $d = 0.001688$, $e = -0.005151$, $f = 0.009607$, $g = -0.006142$ and $T_{ref} = 298.15$. By knowledge of $a_{w,suc}$, the Vogel temperature of sucrose $T_{0,suc}$ can be calculated as[68]:

$$T_{0,suc} = 127.9 \cdot \left(1 + 0.4514\left(1 - a_{w,suc}\right) - 0.5\left(1 - a_{w,suc}\right)^{1.7}\right) \qquad (8)$$

Then, the Vogel temperature of the target substance is given by

$$T_{0,SOA} = T_{0,suc} \cdot \frac{T_{g,SOA}}{T_{g,suc}} \qquad (9)$$

This calculation is performed at equal composition (organic mass fraction) and as a function of $T$ and RH. Supplementary Fig. 8 shows that the humidity dependence of water diffusivity in α-pinene SOA obtained using the semi-empirical estimation scheme is consistent with laboratory measurements[27]. At 298 K, both methods coincide closely over the entire RH range. At lower temperatures, divergence at low RH develops, which is shown here for 260 K. Note, however, that the measurement accuracy is low under these cold and dry conditions. Moreover, such low RH is rare in the troposphere. For example, in the upper troposphere, where the glassy phase state plays a role in affecting ice nucleation processes and thus cloud formation, RH must be at or above ice saturation ($S_{ice} = 1$) before nucleation can occur, corresponding to 72 % RH at 240 K. These differences detected in $D_{H_2O}$ between estimates and laboratory measurements are small compared to the large range caused by the altitude-dependent temperature in the atmosphere.

**Sensitivity studies.** Effects of important parameters including $M$, O:C, $k_{GT}$ and $\kappa$ on $T_g$ predictions are examined by sensitivity simulations. Changes of molar masses by 20% in each volatility bin (as specified in Supplementary Table 2) led to variations of $T_g$ of dry SOA by ~5–10% over most continental areas (Supplementary Fig. 9). The base case O:C ratio was assigned based on the first-generation SOA products[42], which can be considered as the lower end. When the O:C ratio of each volatility bin was increased by 50%, $T_g$ of dry SOA was increased only by ~4%, indicating that the impact of O:C ratio is much smaller than that of molar mass. $k_{GT}$ was varied by $\pm 1$ (refs 5,51,69), which led to changes of $T_g$ within 10%, and more significant impact is found over higher RH regions (for example, ocean and high latitude; Supplementary Fig. 10). The hygroscopicity parameter also exerts an important effect on the prediction over high RH areas, where a variation of $\pm 0.05$ in $\kappa$ led to changes of $T_g$ by ~5–15% (Supplementary Fig. 11).

**Data availability.** The data that support the findings of this study are available from the corresponding author upon reasonable request.

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

## Acknowledgements

This work was funded by the Max Planck Society. M.S. thanks support by the School of Physical Sciences at UC Irvine. Y.L. was supported by the National Natural Science Foundation of China (grant no. 41405121). We thank H.P. Dette for helpful comments.

## Author contributions

M.S., Y.L. and U.P. planned the research. A.P.T., V.A.K., S.N.P. and J.L. provided the global simulations. M.S., Y.L., A.P.T., V.K., T.B. and T.K. analysed the data. M.S. and Y.L. wrote the supplement. M.S., Y.L. and U.P. wrote the manuscript. All authors discussed the results and contributed to manuscript editing.

## Additional information

**Competing interests:** The authors declare no competing financial interests.

