## [Peer Review File · Nature Communications]

Reviewers' Comments:

Reviewer #1 (Remarks to the Author)

The paper presents results from global chemistry model to predict the atmospheric SOA phase state based on estimates of glass transition temperatures T_g . The topic of the paper is interesting and timely and the manuscript is well written. Unfortunately the scientific basis on which the modelling work is built on is vague. First of all, the T_g estimates are based on measurements, which mostly don't present SOA constituents but some other organic model compounds. Then this molecular corridor approach based on pure components is stretched to SOA which consists of thousands of different compounds. The effect of water, which is known plasticizer, is estimated by Gordon-Taylor approach developed for polymers. There is no data in the literature showing that the approach is applicable for SOA. In addition, the diffusion coefficient estimation is based on assumption that sucrose is a good reference for atmospheric SOA. This is an assumption which doesn't have any scientific justification. Taken all these uncertainties into account, I conclude that the paper doesn't merit to be published in Nature Communications.

More detailed comments can be found below:

- 1) Glass transition temperature estimates and parametrization (Fig. 1): all measurements and estimations presented in Fig. 1 are done with single compounds. And as far as I understood from the manuscript, T_g only for one SOA component (3-MBTCA) is measured. More SOA components should be measured and the comparisons shown in Figs. 1 a and b should be shown for SOA components only. How much T_g measured and estimated for the same compound differ? In addition, SOA is a mixture of thousands of organic molecules, hence the T_g of the mixture can vary considerably from single compounds. Based on eq. (1) in page 16 the parametrization defined for single compounds is used to estimate the T_{gi} of each volatility bin consisting of hundreds of compounds.
- 2) There seems to be quite a big deviation in measured and estimated points in figure 1b. At some cases the difference is app. 100 K. In addition, it seems that there is much less data in fig. 1c than in fig. 1b. Why is that?
- 3) T_g of mixture of SOA compounds (i.e. different volatility bins) as well as SOA mixed with water due to the hygroscopic growth was estimated using Gordon-Taylor approach. Gordon-Taylor equation is defined for polymers and is applicable only if there is no strong interactions between the molecules. If there are strong interactions, adduct formation, as well as oligomer formation the approach breaks down (Koop et al., 2011). How applicable it is for SOA?
- 4) The estimation of diffusion coefficients from glass transition temperatures.
 - a. The diffusivity estimation is based on the functional form of VFT equation defined for water molecules in sucrose. In the supplement it is written: "... the diffusivity in the well-known reference substance sucrose" with reference to Zobrist et al., 2011. First of all, there is no papers showing that sucrose would be a good reference for atmospheric SOA. Zobrist et al. (2011) studied water diffusion in sucrose, but there is no comparison measurements done with SOA. If authors claim that sucrose is comparable to SOA they should show it by experimental data.
 - b. Based on equations in SI it seems that the basic assumption behind the diffusivity estimation method is that the dependence of D on water mass fraction is similar for SOA and for sucrose particles. Justification for this assumption should be shown by experimental data.
 - c. How is hygroscopicity κ and density ρ_{org} defined for SOA? There are papers showing that κ can vary a lot depending on the water volume fraction (Wex et al., 2009; Pajunoja et al., 2015)
- 5) In the EMAC modelling work the aerosol composition is defined by using aerosol module ORACLE which has no description for diffusion limitations in particle bulk. Instead the traditional VBS approach assuming equilibrium is used. I think there is a pretty big contradiction between the modelling approach and results showing extremely long characteristic mixing timescales for particles. Also, based on my knowledge the ORACLE doesn't have updated VBS description including ELVOCs and LVOCs.

Koop, T., Bookhold, J., Shiraiwa, M. & Poschl, U. Glass transition and phase state of organic compounds: dependency on molecular properties and implications for secondary organic aerosols in the atmosphere. *Phys. Chem. Chem. Phys.* 13, 19238-19255 (2011).

Pajunoja et al., Adsorptive uptake of water by semisolid secondary organic aerosols in the atmosphere, *Geophys. Res. Lett.*, 42,3063–3068 (2015).

Wex et al. Towards closing the gap between hygroscopic growth and activation for secondary organic aerosol: Part 1 – Evidence from measurements, *Atmos. Chem. Phys.* 9, 3987–3997 (2009).

Zobrist, B. et al. Ultra-slow water diffusion in aqueous sucrose glasses. *Phys. Chem. Chem. Phys.* 13, 3514-3526 (2011).

Reviewer #2 (Remarks to the Author)

The authors present results from global model studies on the phase state distribution of secondary organic aerosol (SOA) particles. This is a topic of high interest since the role of SOA in the climate system, for air quality and for health has by far not quantified yet. Many recent lab studies have shown that SOA particles can exhibit a wide range of phase states (from liquid to viscous to semisolid and solid). The phase state determines many chemical and physical properties and interactions within the atmospheric system.

While the current study is clearly a very explorative one, it is a very welcome first and original contribution for the aerosol community. The manuscript is written clearly and the methods and assumptions sufficiently explained. I recommend publication of this manuscript after some mostly minor corrections as outlined below.

Main comments

1) The glass transition temperature can be predicted within +/- 30 K (l. 69). This range seems very large. Only later it is referred to the sensitivity studies in the Supplemental Information where it is shown that reasonable assumptions for O/C ratio, molar mass etc lead to much smaller variations in predicted Tg. I suggest (i) discussing briefly some of the results of the sensitivity studies in the main part of the manuscript and (ii) relating those to the uncertainty of +/- 30K. What are the uncertainties for the empirical coefficients in Equation (1) (Supplemental Information)?

2) I am aware that ambient measurements of the phase state are very sparse. However, wherever possible (e.g. l. 107ff) comparisons to observations should be made – or it should be stated that they are not available.

Minor comments

l. 43/44: In many models, the phase state of SOA is not clearly defined. They could be also solids as they are not assumed to take up any water.

l. 50: Given that the journal is directed to a broad readership, the 'aerosol effects on climate and air quality' and the associated physical and chemical properties should be briefly specified.

l. 105: I suggest adding 'in agreement with observations' before 'where'

l. 134: Add numbers for the equilibration and activation time scales.

l. 140: Competition between homogeneous and heterogeneous ice formation is only expected in a

narrow temperature range (\sim -38C). This information should be added here.

Table S1: Add units to M and Tg

Technical comments

The copy I received was somewhat confusing since references 1-32 appeared before the acknowledgements etc and figures, followed by 'Methods' and references 33-52.

I. 61: Add reference number [4] or replace author name accordingly.

I. 141: Replace 'mode' by 'modes'

I. 328: Replace 'would be' by 'were'

I. 391: Define VFT before first use.

Reviewer #3 (Remarks to the Author)

Shiraiwa et al. present a 3D numerical modeling study to quantify the phase state of organic aerosol. At the core of the estimate is a new parameterization of the glass transition temperature (Tg) based on the atomic O:C ratio and molecular mass. A correlation between volatility and Tg is noted. The parameterization is applied within a global chemistry climate model. Atmospheric regions of solid, semi-solid, and liquid phase state are identified. Important implications for ice nucleation and transport of pollutants are raised and discussed.

This is a very well written and easily readable manuscript on a hot topic. The new parameterization to predict Tg is useful and a contribution in its own right. The underlying modeling treatments (including O:C ratio, Gordon-Taylor, Boyer-Kauzmann, hygroscopicity parameter, Vogel-Fulcher-Tammann, and Stokes-Einstein) to relate chemical composition, water content, viscosity, and mixing time scales are well established formulations that are currently discussed in the organic aerosol literature. Parameters that were selected to initialize the calculations are somewhat reasonable and in principle supported by experimental data. Sensitivity calculations to small variations in these parameters are included in the supplement. In this sense, the paper is formally correct.

It should be pointed out that much of the framework is 'hypothetical' inasmuch as actual data on realistic complex organic compounds are still not available. For example, I am not aware of Gordon Taylor constants for actual SOA systems. The range in kGT may be much larger than the ± 1 assumed here. The application of $\kappa = 0.1$ is likely wrong. Hygroscopic growth measurements of SOA generated in environmental chambers at RH < 80% often show much smaller κ values. Whether the κ model produces realistic water contents for SOA at temperatures colder than laboratory conditions is unknown. Temperature effects and non-ideal behavior will play important roles. Since phase state is extremely sensitive to small variations in water content I even have fundamental doubts that canonical experimental approaches at room temperature could appropriately resolve the issue. The application of Stokes-Einstein to the diffusivity of water is questionable based on the recent Price et al. paper (<http://www.atmos-chem-phys.net/14/3817/2014/>). Whether it applies at cold temperature is yet another question. The modeling SOA Tg is entirely hypothetical as it is tied to estimated melting points without any data verification. Information about fragility of actual SOA is unknown. Most importantly, the organic aerosol is likely internally mixed. Contributions to aerosol water content for the inorganic species will plasticize the particle and move the composition into a liquid state (and perhaps even a liquid-liquid phase separated state, although that is immaterial to this work). For this, RH history of

the particle needs to be considered as well. In this sense, the predictions of phase state made by this paper are almost certainly wrong.

The limitations raised above are fundamental due to lack of experimental data and fundamental scientific understanding of the subject. I have no doubt that the authors are fully aware of most, if not all of these points.

Despite my critiques, I cautiously recommend the manuscript for publication. Although the numbers will likely undergo substantial revisions in future publications, there is significant value to constrain the regions where solid phase state is a potential concern. This paper represents an acceptable first guess on how these regions may be distributed around the globe.

Response to the comments of Anonymous Referee #1

Referee Comment Summary: The paper presents results from global chemistry model to predict the atmospheric SOA phase state based on estimates of glass transition temperatures T_g . The topic of the paper is interesting and timely and the manuscript is well written. Unfortunately the scientific basis on which the modelling work is built on is vague. First of all, the T_g estimates are based on measurements, which mostly don't present SOA constituents but some other organic model compounds. Then this molecular corridor approach based on pure components is stretched to SOA which consists of thousands of different compounds. The effect of water, which is known plasticizer, is estimated by Gordon-Taylor approach developed for polymers. There is no data in the literature showing that the approach is applicable for SOA. In addition, the diffusion coefficient estimation is based on assumption that sucrose is a good reference for atmospheric SOA. This is an assumption which doesn't have any scientific justification. Taken all these uncertainties into account, I conclude that the paper doesn't merit to be published in Nature Communications.

Response: We thank Anonymous Referee #1 for the review. We do disagree with the comments regarding the scientific justification of our assumptions. First, we would like to point out that the Gordon-Taylor approach has been validated with α -pinene oxidation products (Dette et al., 2014). Second, we did not assume that the water diffusivity in SOA is the same as in sucrose. Instead, the water diffusivity in SOA was estimated applying the Vogel-Fulcher-Tammann (VFT) approach using sucrose as a reference substance (Berkemeier et al., 2014). Diffusion in sucrose has been studied extensively as a function of temperature and relative humidity (Zobrist et al., 2011; Bones et al., 2012). The VFT method has been shown to work well for SOA in previous studies (Berkemeier et al., 2014; Price et al., 2015; Lienhard et al., 2015; Berkemeier et al., 2016). The figure below (it has been added to the supplementary material of the paper as Figure S8) shows the comparison of results from the estimation method in Berkemeier et al. (2014) applied to α -pinene SOA with the laboratory measurements of Lienhard et al., (2015). For this comparison, water diffusivities were calculated using the following parameters for α -pinene SOA at O:C = 0.5, $T_{g,SOA} = 278.5 \pm 7$ K, $\kappa_{org} = 0.12 \pm 0.02$, $k_{GT} = 1.5 \pm 1$, and $\rho_{org} = 1.4$ g cm⁻³. It shows that the humidity

dependence of water diffusivity in α -pinene SOA obtained using the semi-empirical estimation scheme is consistent with laboratory measurements in an electrodynamic balance³. At 298 K, both methods coincide perfectly over the entire relative humidity range. At lower temperatures, divergence at low relative humidity develops, which is showcased here for 260 K. Note however that measurement accuracy is considerably lowered under these very cold and dry conditions. Moreover, such low relative humidity is rare in the troposphere. For example, in the upper troposphere, where the glassy phase state plays a role in affecting ice nucleation processes and thus cloud formation, RH must be at or above ice saturation ($S_{ice} = 1$) before nucleation can occur, corresponding to 72 % RH at 240 K. Note also that these differences detected in D_{H_2O} between estimates and laboratory measurements are small compared to the large range caused by the height-dependent temperature in the atmosphere. Thus, the use of the Gordon-Taylor approach and the water diffusivity estimation by the VFT method both have a solid scientific basis. These issues are now explained in detail in the revised paper.

Figure S8. Comparison of water diffusion coefficients in α -pinene SOA (O:C ratio = 0.54) obtained from water uptake experiments in an electrodynamic balance (Lienhard et al., 2015; green shaded areas) with values obtained with the semi-empirical estimation method developed in Berkemeier et al. (2014) (orange shaded areas) at 260 K and 298 K.

Referee General Comments

1) Glass transition temperature estimates and parametrization (Fig. 1): all measurements and

estimations presented in Fig. 1 are done with single compounds. And as far as I understood from the manuscript, T_g only for one SOA component (3-MBTCA) is measured. More SOA components should be measured and the comparisons shown in Figs. 1 a and b should be shown for SOA components only. How much T_g measured and estimated for the same compound differ? In addition, SOA is a mixture of thousands of organic molecules, hence the T_g of the mixture can vary considerably from single compounds. Based on eq. (1) in page 16 the parametrization defined for single compounds is used to estimate the T_g of each volatility bin consisting of hundreds compounds.

Response: Please note that Fig. 1a includes only SOA compounds. The right figure shows estimated T_g of SOA compounds only as a function of molar mass. Please also note that in this study the “estimated” T_g represents T_g derived from the melting temperature (T_m) following the Boyer-Kauzmann rule (Koop et al., 2011), while the “predicted” T_g is calculated from our new parameterization as a function of molar mass and the O:C ratio. The comparison of measured and estimated T_g shows that T_g can be estimated well by T_m , with a correlation coefficient of 0.97 (Please see below new Fig. S1b). As pointed out, measurements

of T_g for SOA components are scarce (Dette et al., 2014). Thus we validate our parameterization by comparing the predicted T_g with both measured T_g (Koop, et al., 2011; Dette et al., 2014) as well as estimated T_g for 654 SOA components (Shiraiwa et al., 2014) (Fig. 1c). We clarified the above in the revised manuscript and added Fig. S1b to the Supplement.

Figure. Estimated (squares) T_g of SOA compounds plotted against molar mass. The markers are color-coded by molecular O:C ratio.

Figure S1. (a) Measured (circles) and estimated glass transition temperature (T_g) of organic compounds as a function of O:C ratio. Organic compounds with measured T_g are from Koop et al. (2011) and Dette et al. (2014). Those with estimated T_g are 654 SOA components from Shiraiwa et al. (2014). (b) Comparison of measured and estimated T_g for 179 organic compounds (Koop et al., 2011; Dette et al., 2014). The markers are color-coded by molar mass.

We are fully aware that SOA is a mixture of thousands of organic molecules; hence the T_g of the mixture can vary considerably compared to that of single compounds. In current global models, it is computationally prohibitive to simulate individual SOA compounds (even if they would be known individually). The volatility basis set (VBS) framework lumps the thousands of organic compounds into groups (surrogate species) with logarithmically spaced effective saturation concentrations to simulate SOA formation in regional and global models. In this study, the average molar mass and O:C ratio of SOA compounds were assigned to each volatility bin based on molecular corridors. Then, the average T_g for SOA compounds in each volatility bin was estimated using the newly developed parameterization. The T_g of the dry SOA (mixture of SOA compounds for all volatility bins) and wet SOA considering hygroscopic growth was then estimated using the Gordon-Taylor approach, which has been validated in previous studies (see below response to comment 3). As pointed out, it is indeed an approximation to represent complex mixtures of SOA with average T_g , but we consider this a good starting point for global T_g estimation, making use of the successful VBS and

molecular corridors approach. Otherwise, in practice it would not be feasible to perform these calculations in any currently available modeling setup, especially with a global model. This limitation is now discussed in the revised manuscript.

2) There seems to be quite a big deviation in measured and estimated points in figure 1b. At some cases the difference is app. 100 K. In addition, it seems that there is much less data in fig. 1c than in fig. 1b. Why is that?

Response: Please note that Figure 1b does not show a comparison between measured and estimated T_g points, but aims to show side-by-side the dependence of T_g on molar mass and O:C ratio with both measured as well as estimated points representing different compounds; “measured” markers represent organic compounds with measured T_g , and “estimated” markers represent SOA compounds with T_g estimated using the melting point estimation and the Boyer-Kauzmann rule (Koop et al., 2011). There is actually the same number of data points in Fig. 1b and 1c. In Fig. 1c, many of the data points are overlapping with each other near the 1:1 line, which is an indicator for the fact that the inferred correlation works quite well. We clarified this in the main text and the caption of Fig. 1 in the revised manuscript.

3) T_g of mixture of SOA compounds (i.e. different volatility bins) as well as SOA mixed with water due to the hygroscopic growth was estimated using Gordon-Taylor approach. Gordon-Taylor equation is defined for polymers and is applicable only if there are no strong interactions between the molecules. If there are strong interactions, adduct formation, as well as oligomer formation the approach breaks down (Koop et al., 2011). How applicable it is for SOA?

Response: It is correct that the Gordon-Taylor was originally developed for (co)polymers, but it has been applied to and validated for a wide range of mixtures including organic solvent/polymer, water/polymer, molecular organics/molecular organics, as well as water/molecular organics mixtures, see e.g. papers from the food and pharmaceutical sciences (e.g. Roos (1993), Hancock and Zografis (1994) as well as Zobrist et al. (2008) and references therein). For SOA compounds the approach has been validated for the α -pinene oxidation products 3-MBTCA and pinonic acid (Dette et al., 2014) and also for mixtures of oxidized

organic molecules (oligo-alcohols and oligo-carboxylic acids) with inorganic electrolytes (NH_4HSO_4 and NaNO_3) (Dette and Koop, 2015). As pointed out, if there are strong interactions leading to adduct formation, this approach may break down; however, there has been no experimental evidence thus far showing failure of this approach for SOA compounds. Moreover, adduct formation is more likely to occur in binary or ternary mixtures, where the individual components can have high mole fractions. However, adduct formation is highly unlikely in multicomponent mixtures such as SOA with hundreds of compounds, because the individual mole fractions are typically very small. Thus, in multicomponent mixtures particular interactions between individual compounds are more likely to average out (the overall entropic terms overcompensate any strong individual enthalpic interaction terms), thus favoring a mean-field type Gordon-Taylor approach. Note also that the Gordon-Taylor approach is applicable to oligomers, as it does work for polymers. Thus, based on these considerations and on the available experimental SOA data so far, the application of the Gordon-Taylor approach appears to be reasonable. This aspect should be definitely investigated further in follow-up studies. We have added this important point in the revised manuscript.

4) The estimation of diffusion coefficients from glass transition temperatures.

a. The diffusivity estimation is based on the functional form of VFT equation defined for water molecules in sucrose. In the supplement it is written: "... the diffusivity in the well-known reference substance sucrose" with reference to Zobrist et al., 2011. First of all, there are no papers showing that sucrose would be a good reference for atmospheric SOA. Zobrist et al. (2011) studied water diffusion in sucrose, but there is no comparison measurements done with SOA. If authors claim that sucrose is comparable to SOA they should show it by experimental data. **b.** Based on equations in SI it seems that the basic assumption behind the diffusivity estimation method is that the dependence of D on water mass fraction is similar for SOA and for sucrose particles. Justification for this assumption should be shown by experimental data.

Response: Please see our response to the first comment. The VFT method developed in

Berkemeier et al. (2014) has been compared with actual measurements of water diffusivity in α -pinene SOA by Lienhard et al. (2015), showing very good agreement as shown in new Fig. S8.

b. How is hygroscopicity κ and organic density defined for SOA? There are papers showing that κ can vary a lot depending on the water volume fraction (Wex et al., 2009; Pajunoja et al., 2015)

Response: The density of SOA particles is assumed to be 1.4 g cm^{-3} based on previous work (Cross et al., 2007). The hygroscopicity κ is assumed to be 0.1 based on numerous field studies (Gunthe et al., 2009; Roberts et al., 2010; Rose et al., 2011; Wu et al., 2013) and laboratory experiments (Lambe et al., 2011). Model simulations have also suggested a global average of κ of 0.1 (Pringle et al., 2010). A comprehensive laboratory study for determination of sub-saturated κ using a hygroscopicity tandem differential mobility analyzer (HTDMA) suggested that κ of α -pinene, isoprene and TMB SOA are in the range of $\sim 0.04 - 0.16$ depending on oxidation states (Fig. 7 in Duplissy et al., 2011). Thus, a κ of 0.1 appears a reasonable estimate. Nevertheless, as pointed out, κ can vary from this value depending on RH (Pajunoja et al., 2015). To address this sensitivity we have conducted sensitivity simulations with $\kappa = 0.05$ and 0.15 to quantify the effects of κ on the estimation of the glass transition temperature in Fig. S11. We have clarified this point in the revised manuscript.

5) In the EMAC modelling work the aerosol composition is defined by using aerosol module ORACLE which has no description for diffusion limitations in particle bulk. Instead the traditional VBS approach assuming equilibrium is used. I think there is a pretty big contradiction between the modelling approach and results showing extremely long characteristic mixing timescales for particles. Also, based on my knowledge the ORACLE doesn't have updated VBS description including ELVOCs and LVOCs.

Response: This is a good point and there is indeed, at least in principle, a potential inconsistency, which however is expected to have a small effect on our results. Ye et al. (2016) conducted chamber experiments finding that α -pinene SOA mixes relatively quickly above 40% RH at room temperature. Liu et al. (2016) also observed a rapid partitioning of

SOA compounds above 20-30% RH for anthropogenic SOA and for all RH for biogenic SOA. These new studies strongly suggest that kinetic limitations in the bulk do not significantly affect SOA partitioning in the boundary layer, in agreement with our results of T_g at the surface (see Fig. 2, top panel) and justifying our use of equilibrium partitioning in this part of the atmosphere. As shown below in new Fig. S5c, in the current model, most SOA formation (around 95%) occurs in the boundary layer with conditions similar to the above experimental studies. Therefore, the apparent inconsistency may be relevant to only a small fraction (around 5%) of the SOA formation in this modeling study, and does not significantly affect our results, without implications for our conclusions. Nevertheless, a more comprehensive way would be the application of a model such as the kinetic multi-layer model of gas-particle interactions (KM-GAP; Shiraiwa et al., 2012) that resolves particle phase diffusion in a number of chamber experiments to determine aerosol yields, and then treat SOA formation kinetically in the global model. This approach is extremely costly and, thus, is beyond the scope of the current study, but it should be explored in future studies. We clarify these points in the revised manuscript.

The current ORACLE module simulates SOA components with effective saturation concentrations as low as $0.01 \mu\text{g m}^{-3}$ at 298 K for SOA-sv and SOA-iv (SOA from semivolatile and intermediate volatility compounds) and $1 \mu\text{g m}^{-3}$ for SOA-v (SOA from volatile organic compounds). Compounds of even lower volatility (LVOCs and ELVOCs) are not simulated for reasons of computational efficiency and because the predicted OA mass concentration fields are not sensitive to their explicit simulation. Please note that the simulation of these extremely low volatility compounds is mostly relevant for the prediction of number concentrations due to their role in the formation and growth of new particles. Nevertheless, these compounds are implicitly included in the corresponding lower volatility bin of the VBS during the fitting of the corresponding smog chamber results. The accuracy of this simplification has been evaluated in previous applications of ORACLE and the comparisons of its predictions with available field measurements (Tsimpidi et al., 2016). This information is now extensively described in the Methods section of the revised manuscript.

Figure S5c. Modeled mean vertical profiles of concentrations of SOA gas precursors during the years 2005-2009. The simulation grids covered by the Amazon basin, Europe, East China, U.S., India and Sahara are shown in Fig. 2(a).

Response to the comments of Anonymous Referee #2

Referee Comment Summary: The authors present results from global model studies on the phase state distribution of secondary organic aerosol (SOA) particles. This is a topic of high interest since the role of SOA in the climate system, for air quality and for health has by far not quantified yet. Many recent lab studies have shown that SOA particles can exhibit a wide range of phase states (from liquid to viscous to semisolid and solid). The phase state determines many chemical and physical properties and interactions within the atmospheric system. While the current study is clearly a very explorative one, it is a very welcome first and original contribution for the aerosol community. The manuscript is written clearly and the methods and assumptions sufficiently explained. I recommend publication of this manuscript after some mostly minor corrections as outlined below.

Response: We thank Anonymous Referee #2 for the review and the positive evaluation of our manuscript. Our responses to the comments are described below.

Referee Main Comments:

1) The glass transition temperature can be predicted within +/- 30 K (l. 69). This range seems very large. Only later it is referred to the sensitivity studies in the Supplemental Information where it is shown that reasonable assumptions for O/C ratio, molar mass etc lead to much smaller variations in predicted T_g . I suggest (i) discussing briefly some of the results of the sensitivity studies in the main part of the manuscript and (ii) relating those to the uncertainty of +/- 30K. What are the uncertainties for the empirical coefficients in Equation (1) (Supplemental Information)?

Response: We conducted statistical analysis to evaluate the uncertainty of \$T_g\$ prediction. In Fig 1c, we added 68% prediction (dotted lines) and confidence (dashed lines) bands, respectively. The 68% prediction range indicates that that \$T_g\$ of individual SOA compounds can be predicted within \$\pm 20\$ K. Note that the confidence interval is only about \$\pm 3\$ K, which represents the actual uncertainty of multicomponent SOA mixtures under ideal mixing conditions. Our prediction overestimated the \$T_g\$ of phthalates substantially, however these are minor SOA components. We have added this new analysis and clarified these points in the

revised manuscript.

Figure 1. (c) Predicted T_g using a parameterization developed in this study compared to measured (circles) and estimated (squares) T_g with the Boyer-Kauzmann rule. The solid line shows 1:1 line and the dashed and dotted lines show 68% confidence and prediction bands, respectively.

Following the referee’s suggestion, we have moved some of the discussion of the model sensitivity and uncertainty to the main text. More specifically the following text has been added:

“Sensitivity simulations showed that variation of the key parameters including molar mass, hygroscopicity, and the Gordon-Taylor constant led to changes of T_g within 15% (see Supplement).”

The uncertainties for the empirical coefficients in Equation (1) have been added to the Methods section.

2) I am aware that ambient measurements of the phase state are very sparse. However, wherever possible (e.g. l. 107ff) comparisons to observations should be made – or it should be state that they are not available.

Response: This is a helpful suggestion. In addition to the available measurements in the Amazon and Finland, there has been one more field study that has determined the phase state of ambient organic particles. Using scanning X-ray microscopy O’Brien et al. (2014) found

that organic particles collected in California, Mexico City, and Chile have higher viscosities, which is consistent with our model predictions. Moreover, our prediction of solid state under low temperatures is consistent with recent chamber experiments, showing that α -pinene SOA particles exist in a viscous state under low temperatures corresponding to the cirrus region of the free troposphere (Jarvinen et al., 2016). We added these comparisons to the revised manuscript.

Referee Minor Comments:

3) l. 43/44: In many models, the phase state of SOA is not clearly defined. They could be also solids as they are not assumed to take up any water.

Response: We agree that the phase state of SOA is not clearly defined in many air quality models. In most cases, however, air quality / climate models often implicitly assume that SOA consists of liquid particles with condensed phase diffusion rates that are fast enough to maintain equilibrium with the gas phase. Thus, we would prefer to keep this sentence.

4) l. 50: Given that the journal is directed to a broad readership, the ‘aerosol effects on climate and air quality’ and the associated physical and chemical properties should be briefly specified.

Response: We have followed the reviewer’s suggestion and added the following sentence in the revised manuscript:

‘SOA particles influence climate by scattering sunlight and serving as nuclei for cloud droplets and ice crystals.’

5) l. 105: I suggest adding ‘in agreement with observations’ before ‘where’

Response: We have added this point.

6) l. 134: Add numbers for the equilibration and activation time scales.

Response: Characteristic diffusion timescales of water molecules in SOA particles with a diameter of 200 nm are of the order of microseconds near the Earth’s surface and seconds at 850 hPa. We did not model timescales of activation to cloud droplets explicitly in this study,

but inhibition of activation of cloud condensation nuclei is not expected in the absence of kinetic limitations of bulk diffusion of water molecules.

7) l. 140: Competition between homogeneous and heterogeneous ice formation is only expected in a narrow temperature range ($\sim -38^{\circ}\text{C}$). This information should be added here.

Response: We agree with this point. The sentence has been revised to: ‘In that case, glassy states and solid/liquid core-shell morphologies can persist sufficiently long, so that heterogeneous ice nucleation in the deposition and immersion modes may dominate over homogeneous ice nucleation at lower temperatures ($< \sim 240\text{ K}$)¹⁴.’

8) Table S1: Add units to M and T_g

Response: ‘ g mol^{-1} ’ and ‘ K ’ have been added to M and T_g , respectively.

Referee Technical Comments:

9) The copy I received was somewhat confusing since references 1-32 appeared before the acknowledgements etc and figures, followed by ‘Methods’ and references 33-52.

Response: References have been merged. The sequence of ‘Methods’, ‘References’, ‘Acknowledgements’, ‘Author contributions’ and ‘Additional information’ has been adjusted.

10) l. 61: Add reference number [4] or replace author name accordingly.

Response: Reference number [4] has been added.

11) l. 141: Replace ‘mode’ by ‘modes’

Response: It has been corrected.

12) l. 328: Replace ‘would be’ by ‘were’

Response: ‘Would be’ has been replaced by ‘were’.

13) l. 391: Define VFT before first use.

Response: Thanks. 'Vogel-Fulcher-Tammann' is added before 'VFT'.

Response to the comments of Anonymous Referee #3

Shiraiwa et al. present a 3D numerical modeling study to quantify the phase state of organic aerosol. At the core of the estimate is a new parameterization of the glass transition temperature (T_g) based on the atomic O:C ratio and molecular mass. A correlation between volatility and T_g is noted. The parameterization is applied within a global chemistry climate model. Atmospheric regions of solid, semi-solid, and liquid phase state are identified. Important implications for ice nucleation and transport of pollutants are raised and discussed. This is a very well written and easily readable manuscript on a hot topic. The new parameterization to predict T_g is useful and a contribution in its own right. The underlying modeling treatments (including O:C ratio, Gordon-Taylor, Boyer-Kauzmann, hygroscopicity parameter, Vogel-Fulcher-Tammann, and Stokes-Einstein) to relate chemical composition, water content, viscosity, and mixing time scales are well established formulations that are currently discussed in the organic aerosol literature. Parameters that were selected to initialize the calculations are somewhat reasonable and in principle supported by experimental data. Sensitivity calculations to small variations in these parameters are included in the supplement. In this sense, the paper is formally correct.

Responses: We thank the Anonymous Referee #3 for the review and the positive evaluation of our manuscript. Based on the referee's constructive suggestions for improvement, we have revised the manuscript as detailed below.

1) It should be pointed out that much of the framework is 'hypothetical' in as much as actual data on realistic complex organic compounds are still not available. For example, I am not aware of Gordon Taylor constants for actual SOA systems. The range in k_{GT} may be much larger than the +/- 1 assumed here.

Response: The Gordon-Taylor constants k_{GT} for α -pinene oxidation products of 3-MBTCA and pinonic acid have been measured (Dette et al., 2014). As pointed out, k_{GT} for other SOA compounds is still uncertain, but k_{GT} uncertainty of +/- 1 should be a reasonable estimate also based on Koop et al. (2011). We have clarified this point in the revised manuscript.

2) The application of $\kappa = 0.1$ is likely wrong. Hygroscopic growth measurements of SOA generated in environmental chambers at $\text{RH} < 80\%$ often show much much smaller κ values. Whether the κ model produces realistic water contents for SOA at temperatures colder than laboratory conditions is unknown.

Response: We assumed the hygroscopicity of organics κ of 0.1 based on several available field measurements (Gunthe et al, 2009; Roberts et al., 2010; Rose et al., 2011; Wu et al., 2013) and laboratory experiments (Lambe et al., 2011). Model simulations also suggest a global average of κ of 0.1 (Pringle et al., 2010). A comprehensive laboratory study for determination of sub-saturated κ using a hygroscopicity tandem differential mobility analyzer (HTDMA) suggested that κ of α -pinene, isoprene and TMB SOA are in the range of $\sim 0.04 - 0.16$ depending on oxidation states (Fig. 7 in Duplissy et al., 2011). Thus, we think κ of 0.1 is a reasonable estimate. Nevertheless, as pointed out, κ can differ from this value and we have conducted sensitivity simulations with $\kappa = 0.05$ and 0.15 to show the effects of κ on estimation of the glass transition temperature in Fig. S7. We have clarified this point in the revised manuscript.

3. Temperature effects and non-ideal behavior will play important roles. Since phase state is extremely sensitive to small variations in water content I even have fundamental doubts that canonical experimental approaches at room temperature could appropriately resolve the issue. The application of Stokes-Einstein to the diffusivity of water is questionable based on the recent Price et al. paper (<http://www.atmos-chem-phys.net/14/3817/2014/>). Whether it applies at cold temperature is yet another question.

Response: We are aware that the Stokes-Einstein equation is not applicable for small molecules diffusing in a matrix, as we have demonstrated in a previous study (Shiraiwa et al., 2011). Thus, bulk diffusion coefficients of water molecules were estimated, not by the Stokes-Einstein relation, but by the Vogel-Fulcher-Tammann approach as described in the method section. The VFT method has been shown to work well for SOA in previous studies (Berkemeier et al., 2014; Price et al., 2015; Lienhard et al., 2015; Berkemeier et al., 2016). Please also refer to our response to Referee 1. We clarified this point and extended the discussion of the effects of humidity and temperature on diffusivity of water (Fig. S8) in the

Methods section in the revised manuscript.

4. The modeling SOA T_g is entirely hypothetical as it is tied to estimated melting points without any data verification. Information about fragility of actual SOA is unknown. Most importantly, the organic aerosol is likely internally mixed. Contributions to aerosol water content for the inorganic species will plasticize the particle and move the composition into a liquid state (and perhaps even a liquid-liquid phase separated state, although that is immaterial to this work). For this, RH history of the particle needs to be considered as well. In this sense, the predictions of phase state made by this paper are almost certainly wrong. The limitations raised above are fundamental due to lack of experimental data and fundamental scientific understanding of the subject. I have no doubt that the authors are fully aware of most, if not all of these points. Despite my critiques, I cautiously recommend the manuscript for publication. Although the numbers will likely undergo substantial revisions in future publications, there is significant value to constrain the regions where solid phase state is a potential concern. This paper represents an acceptable first guess on how these regions may be distributed around the globe.

Response: We developed the new parameterization of T_g by fitting to the experimental T_g of 179 CH and CHO compounds (Koop et al., 2011; Dette et al., 2014). The estimated T_g based on the Boyer-Kauzmann rule of $T_g = g \cdot T_m$ is only used to validate the new parameterization (Fig. 1c) due to the few measurements available for T_g of SOA components (Dette et al., 2014). We clarified it in the revised manuscript and added Fig. S1b to the Supplement showing that the estimation method of T_g using EPI and the Boyer-Kauzmann rule is adequate. Indeed the fragility of SOA compounds is poorly known and we assumed typical values for organic compounds which are in the range of $\sim 5 - 20$. The associated uncertainty on T_g/T is shown in Fig. S6. We agree that the role of inorganics and associated aerosol water are potentially important and need to be investigated in follow-up studies. The predictions of phase state in this study are consistent with the available measurements in the Amazon, Finland, California, Mexico City, and Chile, which are included in the revised manuscript. Moreover, our prediction of solid state under low temperatures is consistent with recent chamber experiments, showing that α -pinene SOA particles exist in a viscous state under low

temperatures corresponding to the cirrus region of the free troposphere (Jarvinen et al., 2016). We thank again for the encouraging evaluation and we agree that this study should motivate and serve as a starting point for future studies.

References.

- Berkemeier, T., Shiraiwa, M., Pöschl, U., and Koop, T.: Competition between water uptake and ice nucleation by glassy organic aerosol particles, *Atmos. Chem. Phys.*, 14, 12513-12531, 2014.
- Berkemeier, T., Steimer, S., Krieger, U. K., Peter, T., Poschl, U., Ammann, M., and Shiraiwa, M.: Ozone uptake on glassy, semi-solid and liquid organic matter and the role of reactive oxygen intermediates in atmospheric aerosol chemistry, *Phys. Chem. Chem. Phys.*, 18, 12662-12674, 2016.
- Cross, E. S., Slowik, J. G., Davidovits, P., Allan, J. D., Worsnop, D. R., Jayne, J. T., Lewis [†], D. K., Canagaratna, M., and Onasch, T. B.: Laboratory and ambient particle density determinations using light scattering in conjunction with aerosol mass spectrometry, *Aerosol Sci. Technol.*, 41, 343-359, 2007.
- Detle, H. P., Qi, M., Schröder, D. C., Godt, A., and Koop, T.: Glass-forming properties of 3-Methylbutane-1,2,3-tricarboxylic acid and its mixtures with water and pinonic acid, *J. Phys. Chem. A*, 118, 7024-7033, 2014.
- Detle, H. P., and Koop, T.: Glass Formation Processes in Mixed Inorganic/Organic Aerosol Particles, *J. Phys. Chem. A*, 119, 4552-4561, 2015.
- Duplissy, J., DeCarlo, P. F., Dommen, J., Alfarra, M. R., Metzger, A., Barmpadimos, I., Prevot, A. S. H., Weingartner, E., Tritscher, T., Gysel, M., Aiken, A. C., Jimenez, J. L., Canagaratna, M. R., Worsnop, D. R., Collins, D. R., Tomlinson, J., and Baltensperger, U.: Relating hygroscopicity and composition of organic aerosol particulate matter, *Atmos. Chem. Phys.*, 11, 1155-1165, 2011.
- Hancock, B. C. & Zografi, G. The Relationship Between the Glass Transition Temperature and the Water Content of Amorphous Pharmaceutical Solids. *Pharm. Res.* **11**, 471–477 (1994).
- Järvinen, E.; Ignatius, K.; Nichman, L.; Kristensen, T. B.; Fuchs, C.; Hoyle, C. R.; Höppel, N.; Corbin, J. C.; Craven, J.; Duplissy, J.; Ehrhart, S.; El Haddad, I.; Frege, C.; Gordon, H.; Jokinen, T.; Kallinger, P.; Kirkby, J.; Kiselev, A.; Naumann, K. H.; Petäjä, T.; Pinterich, T.; Prevot, A. S. H.; Saathoff, H.; Schiebel, T.; Sengupta, K.; Simon, M.; Slowik, J. G.; Tröstl, J.; Virtanen, A.; Vochezer, P.; Vogt, S.; Wagner, A. C.; Wagner, R.; Williamson, C.; Winkler, P. M.; Yan, C.; Baltensperger, U.; Donahue, N. M.; Flagan, R. C.; Gallagher, M.; Hansel, A.; Kulmala, M.; Stratmann, F.; Worsnop, D. R.; Möhler, O.; Leisner, T.; Schnaiter, M., Observation of viscosity transition in α -pinene secondary organic aerosol. *Atmos. Chem. Phys.* **2016**, 16 (7), 4423-4438.
- Koop, T., Bookhold, J., Shiraiwa, M., and Pöschl, U.: Glass transition and phase state of

organic compounds: dependency on molecular properties and implications for secondary organic aerosols in the atmosphere, *Phys. Chem. Chem. Phys.*, 13, 19238-19255, 2011.

Lienhard, D. M., Huisman, A. J., Krieger, U. K., Rudich, Y., Marcolli, C., Luo, B. P., Bones, D. L., Reid, J. P., Lambe, A. T., Canagaratna, M. R., Davidovits, P., Onasch, T. B., Worsnop, D. R., Steimer, S. S., Koop, T., and Peter, T.: Viscous organic aerosol particles in the upper troposphere: diffusivity-controlled water uptake and ice nucleation?, *Atmos. Chem. Phys.*, 15, 13599-13613, 2015.

Liu, P., Li, Y. J., Wang, Y., Gilles, M. K., Zaveri, R. A., Bertram, A. K., and Martin, S. T.: Lability of secondary organic particulate matter, *Proc. Natl. Acad. Sci. U.S.A.*, doi: 10.1073/pnas.1603138113, 2016.

O'Brien, R. E., Neu, A., Epstein, S. A., MacMillan, A. C., Wang, B., Kelly, S. T., Nizkorodov, S. A., Laskin, A., Moffet, R. C., and Gilles, M. K.: Physical properties of ambient and laboratory-generated secondary organic aerosol, *Geophys. Res. Lett.*, 41, 4347-4353, 2014.

Price, H. C., Mattsson, J., Zhang, Y., Bertram, A. K., Davies, J. F., Grayson, J. W., Martin, S. T., O'Sullivan, D., Reid, J. P., Rickards, A. M. J., and Murray, B. J.: Water diffusion in atmospherically relevant alpha-pinene secondary organic material, *Chem. Sci.*, 6, 4876-4883, 2015.

Pringle, K. J.; Tost, H.; Pozzer, A.; Pöschl, U.; Lelieveld, J., Global distribution of the effective aerosol hygroscopicity parameter for CCN activation. *Atmos. Chem. Phys.* 2010, 10 (12), 5241-5255.

Roos, Y. Melting and glass transitions of low molecular weight carbohydrates. *Carbohydr. Res.* **238**, 39–48 (1993).

Shiraiwa, M., Ammann, M., Koop, T., and Pöschl, U.: Gas uptake and chemical aging of semisolid organic aerosol particles, *Proc. Natl. Acad. Sci. U.S.A.*, 108, 11003-11008, 2011.

Tsimpidi, A. P., Karydis, V. A., Pozzer, A., Pandis, S. N., and Lelieveld, J.: ORACLE (v1.0): module to simulate the organic aerosol composition and evolution in the atmosphere, *Geosci. Model Dev.*, 7, 3153-3172, 2014.

Tsimpidi, A. P., Karydis, V. A., Pandis, S. N., and Lelieveld, J.: Global combustion sources of organic aerosols: model comparison with 84 AMS factor-analysis data sets, *Atmos. Chem. Phys.*, 16, 8939-8962, 2016.

Ye, Q., Robinson, E. S., Ding, X., Ye, P., Sullivan, R. C., and Donahue, N. M.: Mixing of secondary organic aerosols versus relative humidity, *Proc. Natl. Acad. Sci. U.S.A.*, doi: 10.1073/pnas.1604536113, 2016.

Zobrist, B., Marcolli, C., Pedernera, D. A. & Koop, T. Do atmospheric aerosols form glasses? *Atmos. Chem. Phys.* **8**, 5221–5244 (2008).

Reviewers' Comments:

Reviewer #1 (Remarks to the Author)

Authors have adequately addressed to my comments. It's obvious that there is still quite a lot of guessing involved in the work, but I cautiously think that the paper could be acceptable as it represents the first trial of including the SOA phase state description in this kind of modelling frame.

Reviewer #2 (Remarks to the Author)

The authors have addressed all my comments satisfactorily. I recommend publication of the manuscript.

Reviewer #3 (Remarks to the Author)

The authors mostly addressed the comments by the referees, to the point that they can be addressed given the state of the science.

One exception is the assumptions about hygroscopicity. The supersaturated kappa values, modelling kappa values, and measurements at RH = 95% simply do not apply when estimating water content at 30-70% RH. Perhaps the authors can convince themselves by computing kappa at RH = 50% from the measurements by Varutbangkul et al. (<http://www.atmos-chem-phys.net/6/2367/2006/>). Kappa < 0.05 is a much more realistic estimate. Also, as mentioned in the initial review, inclusion of the effect of internally mixed inorganic compounds will drastically change the results. Perhaps the choice of kappa = 0.1 fortuitously accounts for some of these effects.

I stand by my assessment that it is a reasonable first guess study that will likely have to undergo substantial revision in the conclusions, but cautiously recommend publication. However, it is understandable how another referee could arrive at the opposite conclusion.

Response to Referee

Reviewer #3 (Remarks to the Author):

The authors mostly addressed the comments by the referees, to the point that they can be addressed given the state of the science. One exception is the assumptions about hygroscopicity. The supersaturated kappa values, modelling kappa values, and measurements at RH = 95% simply do not apply when estimating water content at 30-70% RH. Perhaps the authors can convince themselves by computing kappa at RH = 50% from the measurements by Varutbangkul et al. (<http://www.atmos-chem-phys.net/6/2367/2006/>). Kappa < 0.05 is a much more realistic estimate. Also, as mentioned in the initial review, inclusion of the effect of internally mixed inorganic compounds will drastically change the results. Perhaps the choice of kappa = 0.1 fortuitously accounts for some of these effects. I stand by my assessment that it is a reasonable first guess study that will likely have to undergo substantial revision in the conclusions, but cautiously recommend publication. However, it is understandable how another referee could arrive at the opposite conclusion.

Response: We thank Referee 3 for positive evaluation of the revised manuscript and an important comment on hygroscopicity. We are aware that hygroscopicity of SOA at moderate RH may be smaller than 0.1. To account for that, we have conducted sensitivity simulation using a hygroscopicity value of 0.05, which would lead to the increase of T_g by 15% at most (Supplementary Figure 11). This information is included in the main text of the revised manuscript. As pointed out, atmospheric SOA particles are often internally mixed with inorganic components, which may push up hygroscopicity. Moreover, a recent study of Pajunoja et al. (2015) has shown that sub-saturated kappa values of biogenic SOA are in the range of 0.05 – 0.15 for 45 -95% RH (Fig. 1). Thus, we think that the use of 0.1 as the base case would be still reasonable. Thanks for pointing out the reference of Varutbangkul et al., which is now included in the revised manuscript.

Reference.

Pajunoja, A., A. T. Lambe, J. Hakala, N. Rastak, M. J. Cummings, J. F. Brogan, L. Q. Hao, M. Paramonov, J. Hong, N. L. Prisle, J. Malila, S. Romakkaniemi, K. E. J. Lehtinen, A. Laaksonen, M. Kulmala, P. Massoli, T. B. Onasch, N. M. Donahue, I. Riipinen, P. Davidovits, D. R. Worsnop, T. Petaja and A. Virtanen; Adsorptive uptake of water by semisolid secondary organic aerosols. *Geophys. Res. Lett.* **42**, 3063-3068 (2015).